# Non-sedating benzodiazepines cause paralysis and tissue damage in the parasitic blood fluke *Schistosoma mansoni*

Paul McCusker[1], Md Yeunus Mian[2], Guanguan Li[2], Michael D. Olp[3], V. V. N. Phani Babu Tiruveedhula[2], Farjana Rashid[2], Lalit Kumar Golani[2], Ranjit S. Verma[4], Brian C. Smith[3], James M. Cook[2], John D. Chan[1]*

1 Department of Cell Biology, Neurobiology & Anatomy, Medical College of Wisconsin, Milwaukee, WI, United States of America, 2 Department of Chemistry and Biochemistry, Milwaukee Institute of Drug Discovery, University of Wisconsin-Milwaukee, Milwaukee, WI, United States of America, 3 Department of Biochemistry, Program in Chemical Biology, Medical College of Wisconsin, Milwaukee, WI, United States of America, 4 Department of Pharmacology and Toxicology, Medical College of Wisconsin, Milwaukee, WI, United States of America

* jdchan@mcw.edu

## Abstract

Parasitic flatworm infections (e.g. tapeworms and fluke worms) are treated by a limited number of drugs. In most cases, control is reliant upon praziquantel (PZQ) monotherapy. However, PZQ is ineffective against sexually immature parasites, and there have also been several concerning reports on cestode and trematode infections with poor PZQ cure-rates, emphasizing the need for alternative therapies to treat these infections. We have revisited a series of benzodiazepines given the anti-schistosomal activity of meclonazepam (MCLZ). MCLZ was discovered in the 1970's but was not brought to market due to dose-limiting sedative side effects. However, in the decades since there have been advances in our understanding of the benzodiazepine GABA$_A$ receptor sub-types that drive sedation and the development of sub-type selective, non-sedating ligands. Additionally, the sequencing of flatworm genomes reveals that parasitic trematodes and cestodes have lost GABA$_A$R-like ligand gated anion channels, indicating that MCLZ's anti-parasitic target is distinct from the human receptors that drive sedation. Therefore, we have screened a library of classical and non-sedating 1,4-benzodiazepines against *Schistosoma mansoni* and identified a series of imidazobenzodiazepines that immobilize worms *in vitro*. One of these hits, Xhe-II-048 also disrupted the parasite tegument, resulting in extensive vacuole formation beneath the apical membrane. The hit compound series identified has a dramatically lower (~1000×) affinity for the human central benzodiazepine binding site and is a promising starting point for the development of novel anti-schistosomal benzodiazepines with minimal host side-effects.

## Author summary

Over 200 million people are infected with schistosomiasis, yet there are limited therapeutic options available to treat this disease. The benzodiazepine meclonazepam is known to

**Data Availability Statement:** All relevant data are within the manuscript and its Supporting Information files.

**Funding:** This work was supported by funds from the Therapeutic Accelerator Program (TAP) and a New Faculty Pilot Grant awarded to J.D.C by the Medical College of Wisconsin. B.C.S was supported by award number R35GM128840 from NIGMS of the National Institutes of Health. M.D.O. is a member of the Medical Scientist Training Program at Medical College of Wisconsin, which is supported in part by National Institutes of Health Training Grant T32GM080202 from the National Institute of General Medical Sciences. J.M.C. was supported by NIMH awards R01MH096463 and R01NS076517, and grant CHE-1625735 from the National Science Foundation, Division of Chemistry. The funders had no role in study design, data collection and analysis, decision to publish, or preparation of the manuscript.

**Competing interests:** The authors have declared that no competing interests exist.

cure both intestinal and urinary schistosomiasis in animal and human studies, but dose-limiting sedation has been a barrier to its development. Little is known about the structure-activity relationship of meclonazepam and other benzodiazepines on schistosomes, or the identity of the parasite receptor for these compounds. However, schistosomes lack obvious homologs to the human GABA$_A$Rs that cause sedation. This indicates that the parasite target of this drug is distinct from the host receptors that underpin dose-limiting side effects of meclonazepam and raises the possibility that benzodiazepines with poor GABA$_A$R affinity may still retain anti-parasitic effects. Here, we report an *in vitro* screen of various benzodiazepines against schistosomes, and the identification of hit compounds that are active against worms yet display reduced affinity for the human GABA$_A$R that causes sedation.

## Introduction

Over 200 million people are infected with the parasitic blood flukes that cause the neglected tropical disease schistosomiasis [1], and over 90% of infections occur in sub-Saharan Africa where the disease kills approximately 280,000 persons/year [2, 3]. Chronic infection adds to the disease burden, putting the socioeconomic cost of schistosomiasis (70 million disability adjusted life years [4]) near HIV/AIDS, malaria or tuberculosis [1]. However, despite these enormous costs treatment relies on just one broad-spectrum drug, praziquantel (PZQ) [5]. PZQ treatment has high cure rates of 70–90% [6, 7], but it is concerning that a subset of infections in human and animal populations appear to be refractory to treatment [8–10], either due to PZQ's lack of efficacy against recently acquired, immature parasites [11, 12] or standing genetic variation in parasite populations. The latter possibility is especially concerning in regard to the potential emergence of PZQ-resistant parasites, and consideration needs to be given to whether PZQ-monotherapy will be sufficient to achieve schistosomiasis elimination [13].

One lead compound with proven anti-schistosomal activity is the benzodiazepine meclonazepam ((S)-3-methylclonazepam, MCLZ). MCLZ was discovered in the 1970's and found to cure both the mature and immature parasites that cause urinary and intestinal forms of schistosomiasis [14]. Development of this lead stalled in the 1980's due to dose-limiting sedation in human trials [15–18]. However, we have re-visited benzodiazepines as potential anti-parasitic leads given advances in two areas. First, it is now understood that the sedative effects of benzodiazepines are driven by GABA$_A$Rs that contain the α1 subunit [19]. This has enabled the design of various benzodiazepines with reduced affinity towards α1-containing GABA$_A$Rs to treat conditions such as asthma and schizophrenia that involve GABA$_A$Rs that contain other α [20]. Second, with recent advances in the sequencing of parasitic helminth genomes there is an abundance of data available to establish whether flatworm parasites possess GABA$_A$Rs [21]. If GABA$_A$Rs are not present in parasitic worms, then it is possible that the structure-activity requirements of anti-parasitic compounds may differ significantly from those mediating benzodiazepine binding to host GABA$_A$Rs, offering the opportunity to develop ligands with increased parasite selectivity. Here, we have profiled the repertoire of *S. mansoni* ligand gated ion channels and, having found no obvious parasite GABA$_A$Rs, screened a library of benzodiazepines to identify compounds that display anti-parasitic activity and exhibit reduced mammalian GABA$_A$R affinity.

## Materials and methods

### Ethics statement

Animal work was carried out with the oversight and approval of the Laboratory Animal Resources facility at the Medical College of Wisconsin, adhering to the humane standards for the health and welfare of animals used for biomedical purposes defined by the Animal Welfare Act and the Health Research Extension Act. Experiments were approved by the Medical College of Wisconsin IACUC committee (approved protocol #AUA00006471 and AUA00006735).

### Bioinformatic prediction of flatworm ligand-gated ion channels

Putative ligand-gated ion channels were curated from a diversity of organisms sampling vertebrates (human), arthropods (*Drosophila melanogaster*), nematodes (*Caenorhabditis elegans*), and mollusks (*Aplysia californica*). The predicted proteomes of these organisms were searched for gene products containing a ligand-gated ion channel (LGIC) Pfam domain (PF02932). These were used to generate hidden Markov probability models (HMMER v3.2.1), which were used to search the predicted proteomes of various free-living *(Schmidtea mediterranea—PRJNA379262, Macrostomum lignano—PRJNA371498)* and parasitic (*Schistosoma mansoni—PRJEA36577, Schistosoma haematobium—PRJNA78265, Clonorchis sinensis—PRJNA386618, Opisthorchis viverrini—PRJNA222628, Echinococcus multilocularis—PRJEB122*) flatworms for putative LGICs. Resulting candidates were filtered based on number of predicted transmembrane domains (TOPCONS predictions of between 3–5 transmembrane domains) [22]. The retained sequences were then aligned (Clustal Omega), manually inspected for the presence of a characteristic Cys loop F/YPxD motif, and degapped (GapStreeze v2.1.0, 25% tolerance) to enable construction of Maximum Likelihood phylogenetic trees (LG model with 500 bootstrap replicates). Homology based searches were also performed to confirm these analyses. TBLASTN searches were performed querying predicted GABA$_A$R protein sequences against the genomic sequences of parasitic flatworms (E-value cut off 1e-5, WormBase ParaSite). Gene product identification numbers for all sequences are provided in S1 Table.

### Chemicals

A complete list of chemical structures is provided in S2 Table. Compounds were either sourced commercially (Toronto Research Chemicals (Meclonazepam) and Sigma Aldrich (clonazepam, nitrazepam, diazepam, bromazepam, flurazepam, lorazepam, flunitrazepam)) or synthesized by the Cook Lab. Compounds synthesized by the Cook lab were chosen for screening on schistosomes based on prior studies identifying GABA$_A$R-sparing benzodiazepines, reported in references [23–26]. Structures were generated in ChemDraw (v17.1) and clustered by physiochemical properties using ChemMine [27]. Detailed synthesis methods for MCLZ analogs MYM-I-88, MYM-I-91A and MYM-II-53 are provided in S1 File.

### Adult schistosome mobility assays

Female Swiss Webster mice infected with *S. mansoni* cercariae (NMRI strain) were sacrificed 49 days post infection by $CO_2$ euthanasia. Adult schistosomes were recovered by dissection of the mesenteric vasculature. Harvested schistosomes were washed in DMEM (ThermoFisher cat. # 11995123) supplemented with HEPES (25mM), 5% v/v heat inactivated FCS (Sigma Aldrich cat. # 12133C) and Penicillin-Streptomycin (100 units/mL). Worms were cultured in 6 well dishes (4–5 male worms in 3mL media per well) in the presence of various test compounds or DMSO vehicle control overnight (37˚C / 5% $CO_2$). Worms were imaged the next

day to record movement phenotypes using a Zeiss Discovery v20 stereomicroscope and a QiCAM 12-bit cooled color CCD camera controlled by Metamorph imaging software (version 7.8.1.0). 1 minute recordings were acquired at 4 frames per second and saved as a .TIFF stack, which was imported into ImageJ for analysis. An outline of the workflow used to quantify movement from these video recordings is shown in S1 Fig. Maximum intensity projections were generated for the entire stack of images (241 frames for a 1-minute recording) and integrated pixel values were measured for the resulting composite image, allowing normalization of movement relative to DMSO control treated worms. Inhibition of movement $IC_{50}$ values were calculated using GraphPad Prism v8.1.1 and are expressed ± 95% confidence intervals. Data represents mean ± standard error for ≥3 independent experiments. Significance (*) was determined by unpaired t-test at a threshold of 0.05.

### Transmission electron microscopy

Adult worms were harvested and recovered as above. Fixation was carried out overnight at 4˚C in 2.5% glutaraldehyde/2% paraformaldehyde in 0.1 M sodium cacodylate (pH 7.3). Worms were washed 3 × 10 minutes in 0.1 M sodium cacodylate and post-fixed for 2 hours on ice in reduced 1% osmium tetraoxide. Worms were then washed 2 × 10 minutes in distilled water and stained overnight at 4˚C in alcoholic Uranyl Acetate. Worms were rinsed in distilled water, dehydrated in 50%, 75% and 95% MeOH, followed by successive 10 minute rinses in 100% MeOH and acetonitrile. Worms were incubated in a 1:1 mix of acetonitrile and epoxy resin for 1 hour prior to 2 × 1 hour incubations in epoxy resin. Worms were then cut transversely and embedded overnight in epoxy resin (60˚C). Ultra-thin sections (70 nm) were cut onto bare 200-mesh copper grids and stained in aqueous lead citrate for 1 minute. Sections were imaged on a Hitachi H-600 electron microscope fitted with a Hamamatsu C4742-95 digital camera) operating at an accelerating voltage of 75 kV.

### Binding assays

Binding assays for benzodiazepines against mammalian GABAARs were performed measuring displacement of [3H] flunitrazepam (0.4 nM) from crude brain membrane preparations [28]. Rat cerebral cortex membrane homogenate (80 µg protein) was incubated with 0.4 nM [3H]-flunitrazepam in 50 mM Tris-HCl (pH 7.7), plus test compounds (screened at concentrations ranging from 0.1 nM to 10 µM), and non-specific binding was assessed by incubation with diazepam (3 µM). Following incubation for 60 min at 4˚C, samples were vacuum filtered through glass fiber filters (GF/B, Packard) presoaked with 0.3% PEI and rinsed with ice-cold 50 mM Tris-HCl (Unifilter, Packard). Filters were dried and counted for radioactivity in a scintillation counter (Topcount, Packard). $K_i$ values of test compounds were calculated by the Cheng-Prusoff equation.

### GABAAR modeling

Benzodiazepines were docked to the human GABAAR cryo-EM structure (PDB ID 6HUO) [29] using Schrödinger Maestro suite v2019-1. Three dimensional ligand structures were generated using the LigPrep module. Water molecules further than 5 Å from the protein surface were removed, hydrogen bonds were optimized at pH 7.0 and the structure was minimized in the OPLS3e forcefield. Ligands were docked into a 10×10×10 Å grid centered on the bound alprazolam molecule in the GABAAR structure using Glide [30] in ExtraPrecision (XP) mode, and output poses were ranked by XP GlideScore. Figures were generated using PyMOL 2.3.0.

## Results

### GABA$_A$Rs are not the schistosome targets of meclonazepam

The benzodiazepine meclonazepam (MCLZ) is an effective anti-schistosomal drug, but the sedative side effects of MCLZ coincide with the anti-parasitic dose [15]. The sedative side effects of MCLZ are likely driven by human GABA$_A$Rs–specifically those heteromeric receptors that contain the α1 subunit [19]. These receptors account for approximately 60% - 80% of brain GABA$_A$Rs [19, 31]. But what is the parasite target of MCLZ? From the earliest reports of anti-schistosomal activity of MCLZ , it has been noted that these effects are not replicated by other benzodiazepines that also bind GABA$_A$Rs with high affinity [14]. Therefore, we considered that the parasite target of MCLZ may be distinct from GABA$_A$Rs, since it is not even clear whether flatworms possess this class of ligand-gated ion channel (LGIC) [32, 33].

In order to comprehensively search the repertoire of flatworm LGICs, we generated hidden Markov probability models (HMMER v3.2.1) using LGICs curated from a diversity of organisms (humans, *D. melanogaster*, *C. elegans*, *A. californica*). These were used to search the predicted proteomes of various free-living (*S. mediterranea*, *M. lignano*) and parasitic (*S. mansoni*, *S. haematobium*, *C. sinensis*, *O. viverrini*, *E. multilocularis*) flatworms to retrieve putative LGICs. Candidates were manually inspected for accurate number of predicted transmembrane domains and the presence of a characteristic Cys-loop F/YPxD motif. The resulting sequences clustered into four groups corresponding to three groups of ligand gated anion channels (Glutamate-gated Chloride Channels (GluCl), GABA$_A$Rs and GABA$_\rho$Rs) and one group of ligand gated cation channel-like gene products (nicotinic acetylcholine receptor (nAchR)-like) (Fig 1A, S1 Table). Previously reported schistosome GluCls [32] and cholinergic receptors [34] clustered alongside sequences consistent with their functional characterization. GABA$_A$Rs are clearly present in the mollusk *A. californica* and free-living flatworms *M. lignano* and *S. mediterranea*, but they appear to have been lost in the parasitic trematode (*S. mansoni*, *S. haematobium*, *C. sinensis*, *O. viverrini*) and cestode (*E. multilocularis*) species analyzed (Fig 1A & 1B). To confirm that GABA$_A$R sequences were not being overlooked due to partial or fragmented gene models, homology-based searches were also performed. TBLASTN search of *A. californica* GABA$_A$Rs against the genomes of parasitic flatworms found to lack GABA$_A$Rs by our HMMER search did not identify additional sequences that clustered with GABA$_A$Rs.

### Non-sedating imidazobenzodiazepines cause parasite contractile paralysis

If schistosomes lack GABA$_A$Rs, then the parasite target of MCLZ may have different structural requirements for ligand binding than the MCLZ:GABA$_A$R interaction that drives sedation. If so, then it should be possible to identify parasite-selective benzodiazepines that retain anti-schistosomal activity but lack affinity for mammalian GABA$_A$Rs that contain α1-subunits. Therefore, we screened a library of compounds that included various α1GABA$_A$R-sparing compounds against adult male *S. mansoni* and assessed action on schistosomes *in vitro*. Worms were harvested from mice 7-weeks post-infection and cultured in test compound (30 μM) overnight, after which video recordings were acquired to measure effects on worm movement relative to vehicle negative control (DMSO 0.1% v/v) and MCLZ (5 μM) positive control. This primary screen of 180 compounds identified 19 ligands that phenocopied MCLZ, exhibiting coiled, contractile phenotype and paralysis (Fig 2A). Active compounds were then re-screened at 10 μM to refine the hits to the most active compounds. This resulted in the prioritization of two chemical series. The first were MCLZ derivatives, including clonazepam (CLZ, lacking the C3 methyl group of MCLZ), and the derivative MYM-I-91A with the phenyl C2' halogen substituted from a chlorine to a fluorine (Fig 2A & 2B). The second series was a

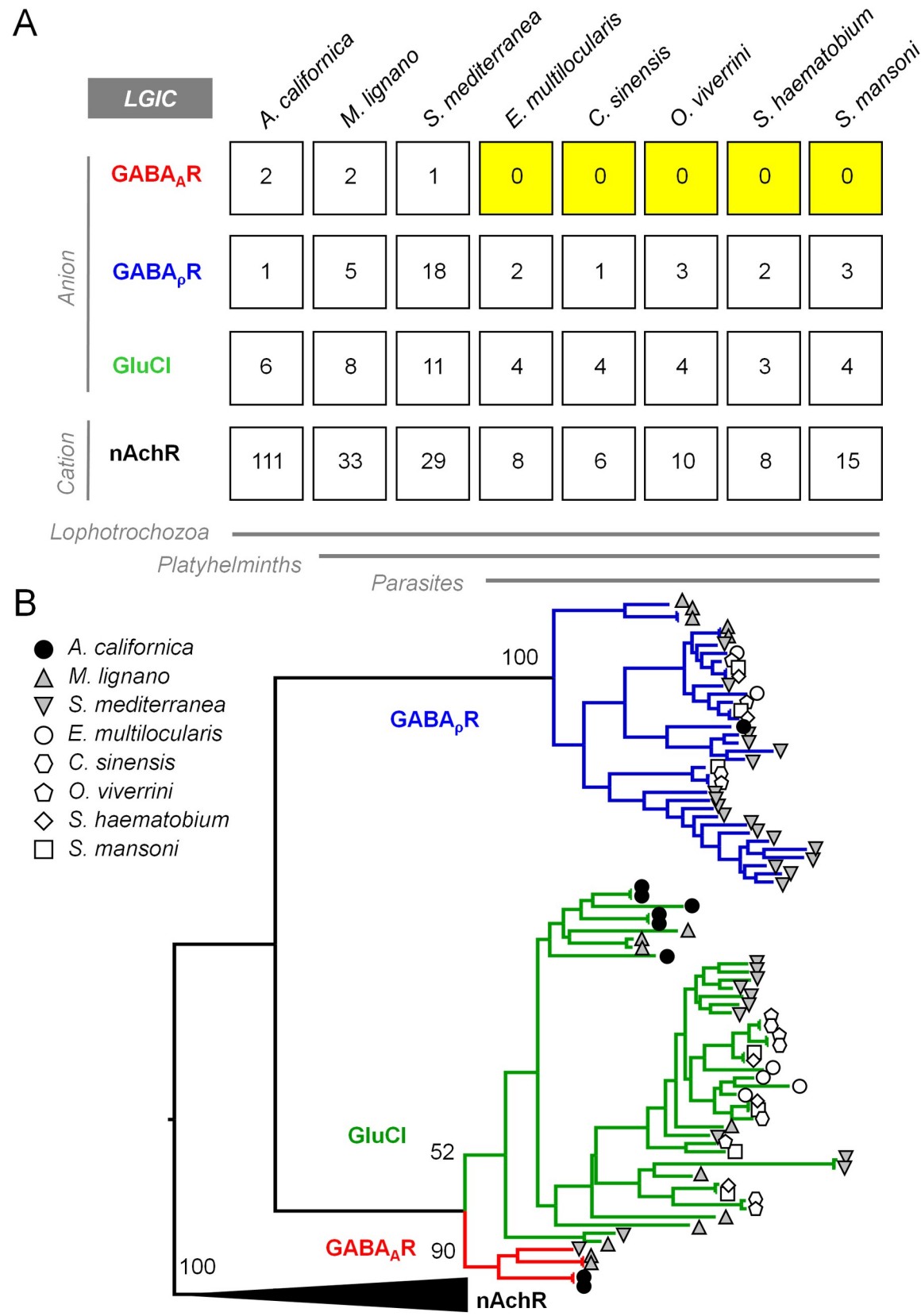

**Fig 1. Comparison of free-living and parasitic flatworm Cys-Loop Ligand Gated Ion Channels (LGICs). (A)** Cys-loop LGICs were curated from the genomes of a non-flatworm lophotrochozoan (*A. californica*) as well as free-living (*M. lignano, S. mediterranea*) and parasitic flatworms (*E. multilocularis, C. sinensis, O. viverrini, S. haematobium, S. mansoni*) and clustered into various families of GABA_AR-like, GABA_ρR-like, GluCl-like and nAChR-like sequences. For gene ID numbers, see S1 Table. **(B)** Phylogeny of flatworm and lophotrochozoan LGICs, with a clade of GABA_AR subunits (red), GABA_ρR subunits (blue) and GluCl subunits (green) that include a previously reported flatworm-specific group of receptors [32]. Bootstrap percentage is shown at nodes (500 replicates).

group of imidazobenzodiazepines (SH-I-055, XliHeII-048 and XHE-II-048) that also varied in the C3 and phenyl C2' positions (Fig 2A & 2C).

The structure activity relationship of these two series was explored, assessing the comparative potency of various derivatives at impairing schistosome movement across a series of doses. Qualitative observations on the activity of various MCLZ-like benzodiazepines have been previously reported [14, 35, 36], and the activity of the MCLZ-like series was largely consistent with these studies. The rank order of potency was MCLZ ($IC_{50}$ 160 nM), CLZ ($IC_{50}$ 1.9 μM), MYM-I-91A ($IC_{50}$ 7.2 μM), MYM-I-88 ($IC_{50}$ 26.0 μM), followed by MYM-II-53 and nitrazepam which were inactive at concentrations as high as 50 μM (Fig 2B). The structure-activity relationship of the imidazobenzodiazepines was explored, assessing the ability of the three hit compounds and structurally related, less active compounds to evoke coiled, contractile paralysis (Fig 2C). The most active compounds, XliHeII-048 ($IC_{50}$ 540 nM) and Xhe-II-048 ($IC_{50}$ 850 nM), contain an imidazole-ester group at the benzodiazepine N1-C2 position and a trimethylsilyl (TMS) acetylene group at the C7 position. The two compounds differ only by the addition of a fluorine at the phenyl C2' position on XliHeII-048. A third compound, SH-I-055 ($IC_{50}$ 1.4 μM) was identical to XliHeII-048 except for the addition of a chiral (S)-methyl group at the C3 position. When this chiral methyl group was in the (R) orientation there was a marked decrease in potency (compound SH-I-060). Finally, compounds retaining SH-I-055's (S)-methyl group but with varying C7 modifications in place of the TMS acetylene group all showed dramatically decreased affinity (GL-I-78, SH-I-48B and SH-053-2'F-S-CH3 with a cyclopropyl, bromine and alkyne group, respectively).

The potency of XliHeII-048 and Xhe-II-048 at inhibiting worm movement was comparable to the active ligands in the MCLZ derivative series with $IC_{50}$ values in the high nanomolar range (Fig 2B & 2C). Therefore, binding assays were performed to compare the relative affinities of these two chemical series for mammalian GABA_ARs (Fig 3). As expected, MCLZ potently displaced [3H]-flunitrazepam from rat brain membrane preparations ($K_i$ = 2.4 nM). The related compound CLZ bound GABA_ARs with an even higher affinity ($K_i$ = 0.82 nM). Analogs MYM-I-91A and MYM-II-53 also displayed high affinity ($K_i$ = 12.1 nM and 4.1 nM, respectively). MYM-I-88, which lacks a halogen on the phenyl ring, displayed markedly reduced binding ($K_i$ = 76.4 nM). The imidazobenzodiazepine series displayed a GABA_AR affinity roughly three logs less potent than MCLZ (Xhe-II-048 $K_i$ = 2.5 μM, SH-I-055 $K_i$ = 1.7 μM, XliHeII-048 $K_i$ = 1.6 μM).

## Imidazobenzodiazepine Xhe-II-048 causes structural damage to parasite tissue

Given the parasite-selectivity of imidazobenzodiazepines (Fig 3, blue) relative to MCLZ-like compounds (Fig 3, red), we investigated the effects of XliHeII-048, Xhe-II-048 and SH-I-055 on schistosome tissues in more detail. Specifically, we were interested in drug-evoked damage to the parasite tegument, which is a feature of many anti-schistosomal compounds [38]. Worms treated with DMSO vehicle control, MCLZ (5 μM) or various imidazobenzodiazepines (10 μM, based on effective concentrations determined in Fig 2) overnight were fixed and processed for imaging by transmission electron microcopy (TEM). Imaging transverse

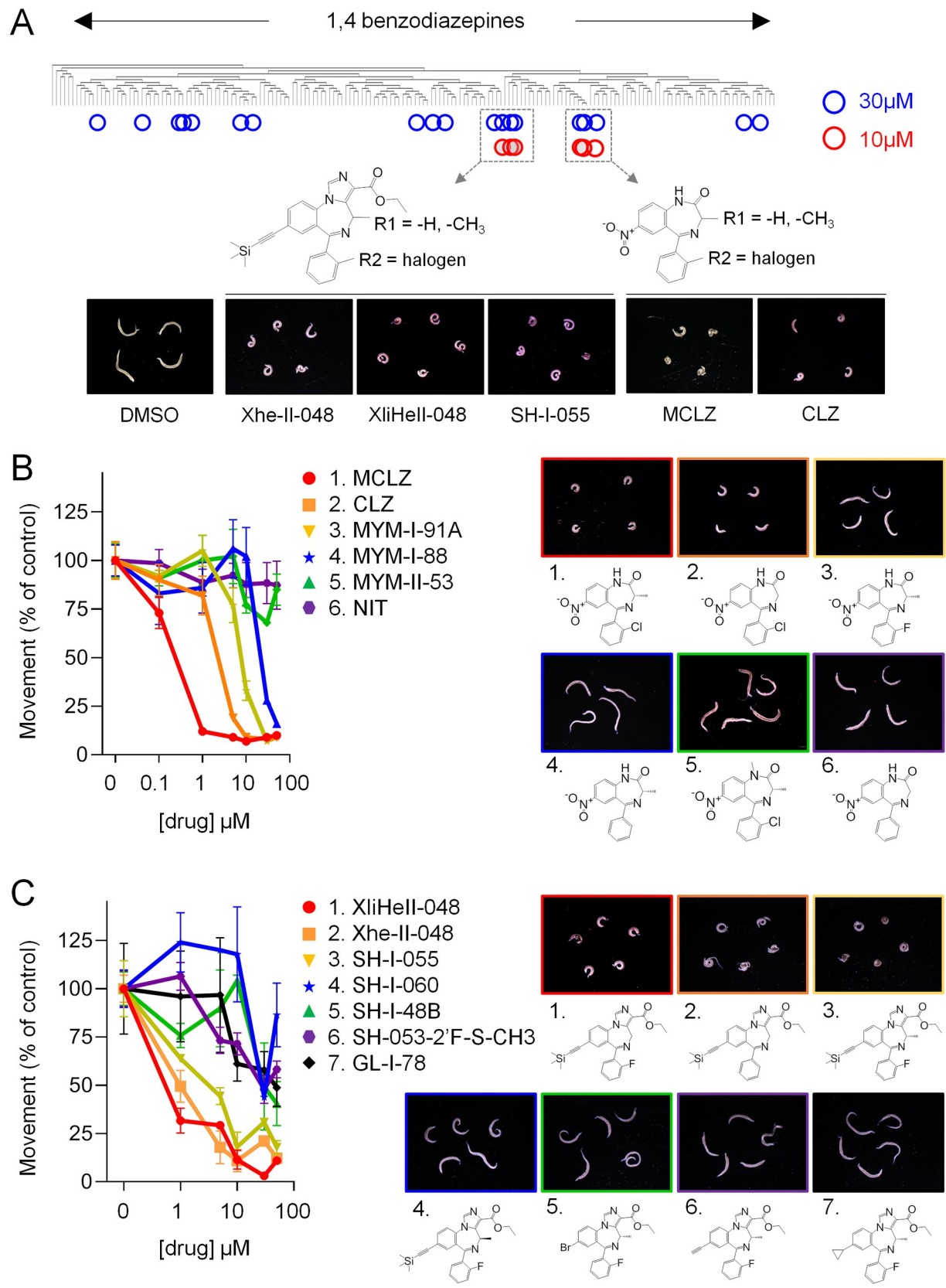

**Fig 2. Identification of benzodiazepines with *in vitro* activity against *S. mansoni*. (A)** 180 benzodiazepines were screened for ability to contract and paralyze schistosomes *in vitro*. Compounds were initially screened at 30 μM (hits = blue circles), and active compounds were re-screened at 10 μM (hits = red circles). This prioritized imidazobenzodiazepines with a TMS-acetylene moiety (left) and meclonazepam-like compounds (right). Structure-activity relationship of **(B)** a series of MCLZ derivatives and **(C)** a series of imidazobenzodiazepines. MCLZ = meclonazepam, CLZ = clonazepam, NIT = nitrazepam. Left = movement concentration-response curves for parasites exposed to each compound. Right = images of drug treated worms (10 μM, overnight).

schistosome cross sections revealed a typical body wall structure in DMSO treated worms, with alternating layers of schistosome muscle, followed by the tegument basal membrane, tegument syncytium, and tegument apical membrane. In MCLZ treated worms, tissue layers are disrupted, with pervasive vacuolization of the tegument (Fig 4A). The tegument of Xhe-II-048 treated worms displayed a similar pattern of extensive vacuole distribution beneath the apical membrane, while worms treated with XliHeII-048 and the less potent imidazobenzodiazepine SH-I-055 displayed normal tissue ultrastructure (Fig 4A).

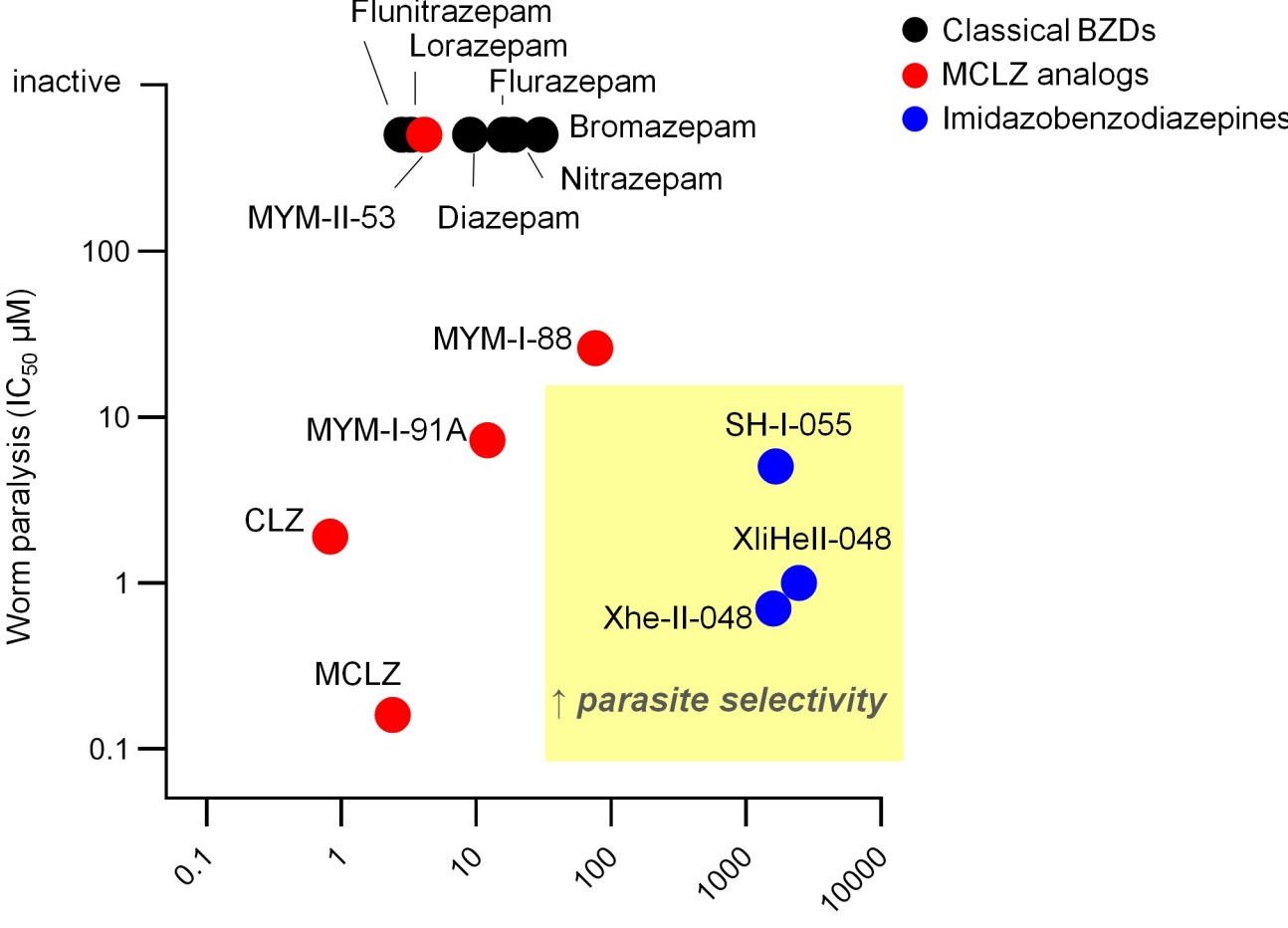

**Fig 3. Relative mammalian GABAAR affinity and schistosome potency of various benzodiazepines.** Scatter plot of mammalian central benzodiazepine binding site affinity ($K_i$) versus schistosome activity (movement IC$_{50}$). Sedating compounds active against worms (i.e. MCLZ) fall within the lower left quadrant. Desired compounds that lack sedation but retain anti-parasitic effects fall within the lower right quadrant. Red = MCLZ analogs. Blue = imidazobenzodiazepines. Black = classical benzodiazepines. Mammalian central benzodiazepine binding site $K_i$ values for flunitrazepam, lorazepam, diazepam, flurazepam, nitrazepam, and bromazepam are from reference [37].

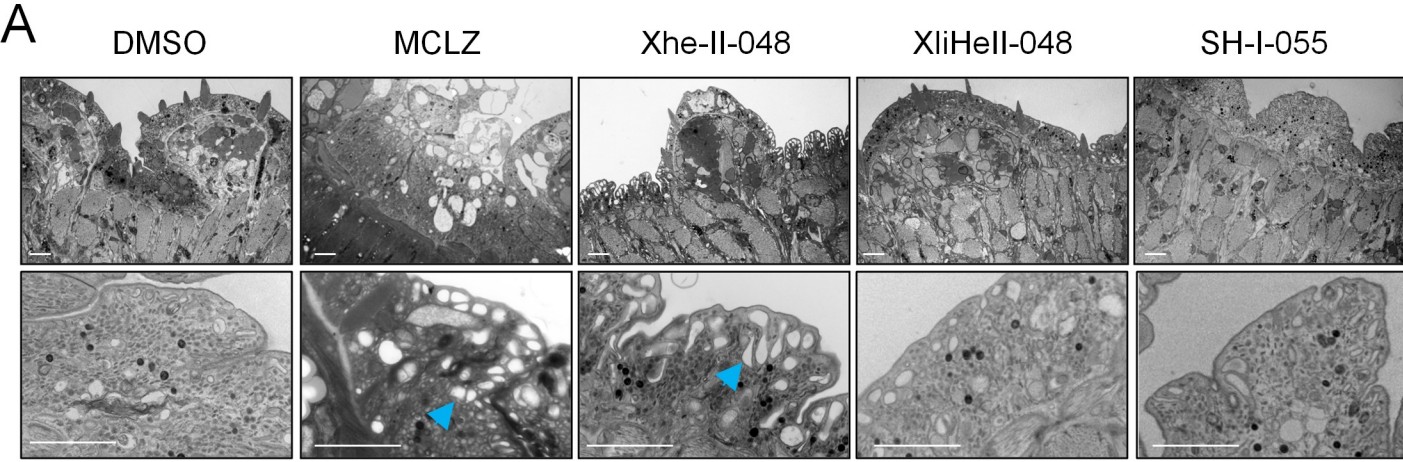

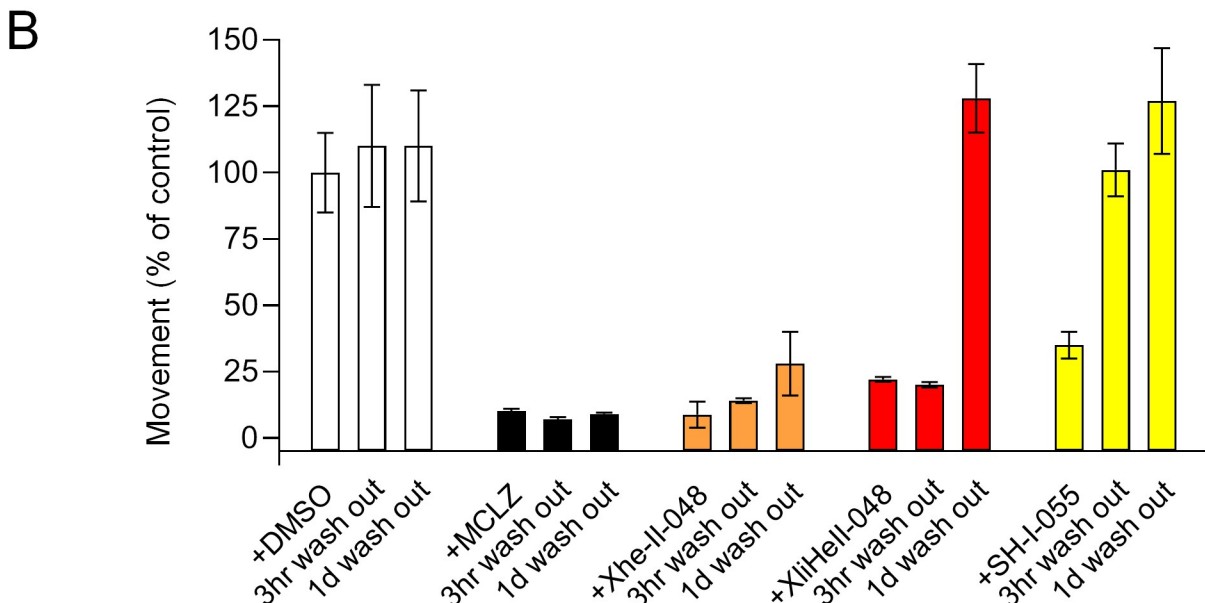

**Fig 4. Xhe-II-048 damages the schistosome tegument. (A)** Transmission electron microscopy images of transverse sections of *S. mansoni* exposed to either DMSO control, MCLZ (5 μM) or various imidazobenzodiazepines (10 μM, 14 hours). Dorsal tegument is oriented to the top. Scale = 2 μm. Arrowed = vacuoles beneath the tegument apical membrane. **(B)** Movement of worms treated with DMSO, MCLZ, or TMS-acetylene imidazobenzodiazepine compounds (10 μM, 14 hours), and recovery at various timepoints (3 hours, 1 day) following drug washout.

MCLZ is a schistocidal compound, and worms do not recover movement following drug washout (Fig 4B). Similarly, Xhe-II-048 treated worms did not recover movement up to one day following drug washout, consistent with the extensive ultrastructural damage caused by this compound (Fig 4A). Compounds XliHeII-048 and SH-I-055, which did not cause pervasive tegument damage, evoked only a transient paralysis, recovering by 1 day after drug washout.

## Discussion

The anti-schistosomal activity of the benzodiazepine meclonazepam (MCLZ) was discovered in the 1970's, but development of this lead as a human therapy for schistosomiasis stalled due

to sedative side effects. While MCLZ was capable of curing infection, the doses required coincided with the onset of sedative side effects [14, 15]. This is expected, given the structural similarity of MCLZ to centrally acting benzodiazepines such as CLZ and NIT that are clinically used as anxiolytics and display well known sedating effects. Some attempts were made in the 1980's to antagonize the sedative effects of MCLZ using the GABA$_A$R antagonist flumazenil. While flumazenil did not impair the anti-schistosomal effect of MCLZ [39], pharmacokinetic differences between flumazenil (administered by IV due to poor bioavailability, <1 hour elimination half-life [40]) and MCLZ (orally bioavailable, half-life up to 80 hours[41]) precluded the development of an admixture as a viable non-sedating therapeutic approach [16, 42]. Consequently, research on MCLZ as an anti-schistosomal lead has slowed over the past several decades. However, we have revisited this compound based on recent helminth genomic data indicating that parasitic flatworms lack GABA$_A$Rs (Fig 1), advances in our understanding of mammalian GABA$_A$R subtypes that account for sedative side effects [19, 43], and advances in the development of non-sedating benzodiazepines with selectivity towards various GABA$_A$R sub-types [23–26].

## Schistosome genomes lack GABA$_A$Rs

Sequenced genomes of parasitic trematode (*S. mansoni*, *S. haematobium*, *C. sinensis*, *O. viverrini*) and cestode (*E. multilocularis*) flatworms lack GABA$_A$Rs, the benzodiazepine targets that cause sedation (Fig 1). Gene loss is common with the evolution of parasitism [44, 45], and a lack of GABA$_A$Rs is consistent with prior bioinformatic characterization of parasitic flatworm LGICs [21, 33, 46].

If parasitic flatworms lack GABA$_A$Rs, might MCLZ act on related cys-loop ligand-gated ion channels (GluCls and GABAρRs)? Experimental and bioinformatic evidence indicates that this is unlikely. MCLZ is inactive against SmGluCl, a representative of the class of flatworm chloride channels most similar to GABA$_A$Rs [32]. Recent structural data resolving the interactions of classical benzodiazepines with human GABA$_A$Rs provide an explanation for this [29]. Specifically, the human α1His102 residue that interacts with the benzodiazepine C7 position at the interface of the GABA$_A$R α1 and γ subunits is replaced by an arginine in the flatworm-specific GluCls. This position is important, since human α4 and α6 GABA$_A$R subunits also contain an arginine in this position, and the larger sidechain likely sterically clashes with classical benzodiazepines to render GABA$_A$Rs comprised of α4 and α6 subunits benzodiazepine-insensitive. In the case of each of the three schistosome GABAρRs, the α1His102 position contains a negatively charged aspartic acid. This switch from a positively charged histidine sidechain may oppose interactions with the electron-dense benzodiazepine C7 position. The inactivity of benzodiazepines on flatworm cys-loop LGICs is consistent with the observation that, aside from MCLZ, and to a lesser degree the structurally similar compound CLZ, benzodiazepines lack anti-schistosomal activity ([14], Fig 3, S2 Table). These findings support the hypothesis that the parasite target is distinct from the human GABA$_A$Rs that account for dose-limiting sedation. However, the possibility that the parasite receptor of MCLZ may have structural similarity to mammalian GABA$_A$Rs (even if it lacks sequence similarity) cannot be excluded until this target is deorphanized.

Resolution of the schistosome target of MCLZ is a high priority, as this may enable the design of ligands with broader anti-parasitic activity. While MCLZ is active against the two major African species of schistosomes, it is inactive against the Asian schistosome *S. japonicum*. This situation is similar to the example of the anti-schistosomal drug oxamniquine. Oxamniquine is only effective against *S. mansoni*, but discovery of the parasite drug target has

ultimately enabled the design of analogs with broad range activity against all three of the major schistosome species (*S. mansoni*, *S. japonicum* and *S. haematobium*)[47, 48].

## *S. mansoni* are paralyzed by a class of α1GABA$_A$R-sparing benzodiazepines

While benzodiazepines are typically considered GABA$_A$R ligands with anxiolytic or sedative properties, numerous α1GABA$_A$R-sparing members of this class have been developed with indications as diverse as anti-asthma to anti-viral medications [25, 49, 50]. Several imidazo-benzodiazepines with a C7 trimethylsilyl (TMS) acetylene group caused schistosome contractile paralysis *in vitro* at high nanomolar to low micromolar concentrations (Fig 2A & 2C) and display low GABA$_A$R binding affinity (Fig 3). These compounds were originally synthesized as part of a chemical series exploring α2 / α3-selective benzodiazepines as potential anxiolytic, anticonvulsant and antinociceptive leads [24, 26]. The differing GABA$_A$R affinities of MCLZ and the imidazobenzodiazepine hits may be due to the interaction of the MCLZ N1 position with the α1 S205, which is disrupted by the addition of the imidazole ring. Modification of this N1 is also observed in the non-sedating antiviral benzodiazepine BDAA [50], although smaller alkyl groups are likely tolerated, such as the methyl on diazepam and MYM-II-53. Additionally, the large TMS-acetylene group at the C7 position of the imidazobenzodiazepine hit series may not be tolerated within the GABA$_A$R binding pocket, where the MCLZ C7 nitro group is predicted to interact with the γ N60 sidechain (Fig 5).

While these ligands phenocopy MCLZ to a degree, it is unclear whether they act via the same schistosome receptor—or if they do bind the same receptor, whether they share a common binding pose. The target of MCLZ will need to be identified to generate hypotheses into ligand-receptor structure-activity relationships (SAR), as the structures of MCLZ and the TMS-acetylene imidazobenzodiazepines appear quite different. However, there are similarities and differences in the SAR of the two series.

Two interesting positions are (i) the benzodiazepine C3 position, which is typically unmodified in classical benzodiazepines but contains a chiral methyl group in these two series, and (ii) the phenyl C2' position, which is commonly halogenated in benzodiazepines with

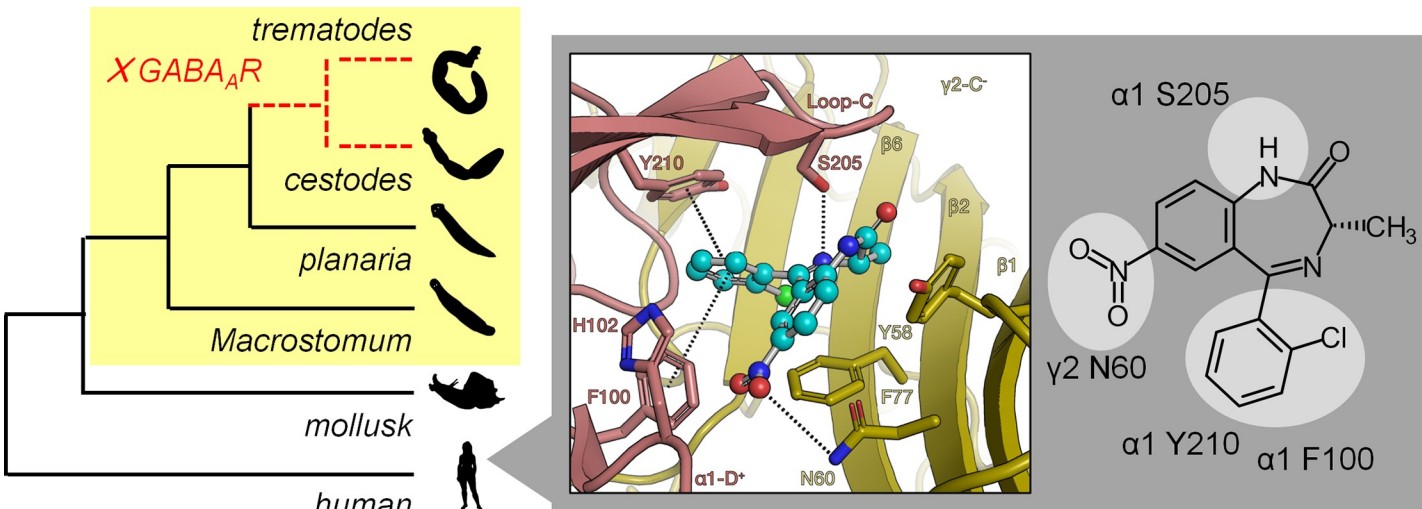

**Fig 5. MCLZ interaction with host GABA$_A$Rs.** Within the flatworm phylum (highlighted yellow), GABA$_A$Rs are present in free living Macrostomum and planarians, but are lost in parasitic flatworms (dashed line). *Inset*—While the flatworm target has still to be identified, *in silico* docking of MCLZ to the benzodiazepine binding site of the solved human heteromeric α1β3γ2 GABA$_A$R cryo-EM structure [29] reveals predicted interactions with amino acid side chains on the α1 and γ2 subunits (dashed lines). *Right* - 2D chemical structure of MCLZ, with potential GABA$_A$R interactions highlighted.

GABA$_A$R affinity. The benzodiazepine C3 position is essential for MCLZ activity, the key difference between clonazepam and MCLZ is that clonazepam lacks a C3 methyl group. This results in a roughly >3 log decrease in potency (Fig 3, [51]). Chirality of this position is also important for 3-methylclonazepam, since the (R) enantiomer reportedly exhibits reduced anti-parasitic efficacy [36]. On the other hand, C3 methylation decreased activity of XliHeII-048. The (S)-methylated compound SH-I-055 was ~3 times less potent than XliHeII-048, and the (R)-methylated compound SH-I-060 was essentially inactive. This schistosome SAR is distinct from the binding of imidazobenzodiazepines to mammalian GABA$_A$Rs, where (S) and (R) isomers have roughly equivalent binding affinities [52]. The halogen on the phenyl C2' position of MCLZ seems important for schistosome activity, given that nitrazepam (inactive against parasites) differs from clonazepam (movement IC$_{50}$ 1.9 μM) in that the phenyl ring is unsubstituted. However, XliHeII-048 (which contains a fluorine at this position) and Xhe-II-048 (which possesses an unsubstituted phenyl ring) appear equipotent (Fig 2C). In fact, the unsubstituted compound Xhe-II-048 was unique in evoking structural damage to the parasite tegument (Fig 4A).

Development of the Xhe-II-048 hit compound into a bona fide anti-schistosomal lead will likely require modifications to improve metabolic stability *in vivo*. Specifically, the ester and TMS-acetylene groups will likely require substitution with bioisosteres. The imidazole ester group is likely rapidly hydrolyzed by carboxylesterases during first pass metabolism, and the TMS-acetylene is also unlikely to be stable *in vivo* [53]. From the SAR shown in Fig 2C, it is apparent that loss of the TMS-acetylene group dramatically decreased potency. Nevertheless, this is the first report of non-sedating benzodiazepines screened against schistosomes. MCLZ has anti-schistosomal activity in human clinical trials, but with an extremely narrow therapeutic index; the effective anti-parasitic dose (0.2–0.3 mg/kg) coincided with the dose at which sedation was reported (above 0.3 mg/kg) [15]. Here, we have identified benzodiazepine hits that exhibit potent anti-schistosomal effects *in vitro* and dramatically lower affinity for host α1GABA$_A$Rs (Fig 3). Given the scarcity of new lead compounds to treat schistosomiasis, these data are valuable in advancing a pharmacophore that retains anti-schistosomal activity while displaying reduced sedation.

## Supporting information

**S1 Fig. Quantification of schistosome movement.** Worm movement was quantified from video recordings (1 minute duration, 4 frames per second). **(A)** Video recordings in color (i) were converted to gray scale and inverted so that worms were transformed to dark silhouettes against a light background (ii). Video recordings (.tiff stacks of 241 images) were treated as a Z-stack, with a composite image of the maximum intensity from each frame integrated into one composite image (iii). **(B)** Movement was quantified by calculating the pixel intensity values of the drug treated composite and expressed relative to the DMSO vehicle control treated composite, producing a numerical quantification of movement across each concentration.
(TIF)

**S1 Table. List of putative Aplysia and flatworm LGICs.** Sequence IDs reflect putative LGICs curated from *A. californica* (all NCBI deposited proteins for taxonomy ID 6500), *S. mediterranea* (assembly ASM260089v1), *M. lignano* (assembly Mlig_3_7), *E. multilocularis* (assembly EMULTI002), *Clonorchis sinensis* (assembly ASM360417v1), *O. viverrini* (assembly Opi-Viv1.0), *S. haematobium* (assembly SchHae_1.0) and *S. mansoni* (assembly v7) and clustered into either nAch, GluCl, GABA$_A$ or GABA$_\rho$ like receptors. Other than *A. californica*, all assemblies were accessed via WormBase ParaSite.
(XLSX)

**S2 Table. Structures and phenotypes of benzodiazepines screened against *S. mansoni*.** Data from the primary screen shown in Fig 2A. SMILES IDs are provided for all compounds screened, and phenotypes are shown for worms exposed to 30 μM test compound overnight. Compounds highlighted in blue active at 30 μM and in red at 10 μM.
(XLSX)

**S1 File. Detailed methods for the synthesis of MCLZ analogs.** Methods for the synthesis of MCLZ derivatives with modifications to the N1 position and phenyl C2' position.
(DOCX)

## Acknowledgments

We thank the Milwaukee Institute for Drug Discovery and University of Wisconsin-Milwaukee's Shimadzu Laboratory for Advanced and Applied Analytical Chemistry for help with spectroscopy. Schistosome-infected mice were provided by the NIAID Schistosomiasis Resource Center at the Biomedical Research Institute (Rockville, MD) through NIH-NIAID Contract HHSN272201000005I for distribution via BEI Resources.

## Author Contributions

**Conceptualization:** Paul McCusker, Md Yeunus Mian, Guanguan Li, Michael D. Olp, V. V. N. Phani Babu Tiruveedhula, Farjana Rashid, Lalit Kumar Golani, Ranjit S. Verma, Brian C. Smith, James M. Cook, John D. Chan.

**Data curation:** Paul McCusker, Md Yeunus Mian, Guanguan Li, Michael D. Olp, V. V. N. Phani Babu Tiruveedhula, Farjana Rashid, Lalit Kumar Golani, Brian C. Smith, John D. Chan.

**Formal analysis:** Paul McCusker, John D. Chan.

**Funding acquisition:** John D. Chan.

**Investigation:** Paul McCusker, Md Yeunus Mian, Guanguan Li, Michael D. Olp, V. V. N. Phani Babu Tiruveedhula, Farjana Rashid, Lalit Kumar Golani, Brian C. Smith, John D. Chan.

**Methodology:** Paul McCusker, John D. Chan.

**Project administration:** John D. Chan.

**Resources:** John D. Chan.

**Software:** Brian C. Smith, John D. Chan.

**Supervision:** V. V. N. Phani Babu Tiruveedhula, Brian C. Smith, James M. Cook, John D. Chan.

**Validation:** John D. Chan.

**Visualization:** Michael D. Olp, Brian C. Smith, John D. Chan.

**Writing – original draft:** John D. Chan.

**Writing – review & editing:** Paul McCusker, Md Yeunus Mian, Guanguan Li, Michael D. Olp, V. V. N. Phani Babu Tiruveedhula, Ranjit S. Verma, Brian C. Smith, James M. Cook, John D. Chan.

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
