## [Decision Letter · Decision Letter 0]

20 Sep 2019

Dear Dr. Chan:

Thank you very much for submitting your manuscript "Non-sedating benzodiazepines cause paralysis and tissue damage in the parasitic blood fluke Schistosoma mansoni " (PNTD-D-19-01242) for review by PLOS Neglected Tropical Diseases. Your manuscript was fully evaluated at the editorial level and by independent peer reviewers. The reviewers appreciated the attention to an important topic but identified some aspects of the manuscript that should be improved.

We therefore ask you to modify the manuscript according to the review recommendations before we can consider your manuscript for acceptance. Your revisions should address the specific points made by each reviewer.

(1) A letter containing a detailed list of your responses to the review comments and a description of the changes you have made in the manuscript.

(2) Two versions of the manuscript: one with either highlights or tracked changes denoting where the text has been changed (uploaded as a "Revised Article with Changes Highlighted" file ); the other a clean version (uploaded as the article file).

(3) If available, a striking still image (a new image if one is available or an existing one from within your manuscript). If your manuscript is accepted for publication, this image may be featured on our website. Images should ideally be high resolution, eye-catching, single panel images; where one is available, please use 'add file' at the time of resubmission and select 'striking image' as the file type. 

Please provide a short caption, including credits, uploaded as a separate "Other" file. If your image is from someone other than yourself, please ensure that the artist has read and agreed to the terms and conditions of the Creative Commons Attribution License at http://journals.plos.org/plosntds/s/content-license (NOTE: we cannot publish copyrighted images). 

(4) Appropriate Figure Files 

Please remove all name and figure # text from your figure files upon submitting your revision. Please also take this time to check that your figures are of high resolution, which will improve both the editorial review process and help expedite your manuscript's publication should it be accepted. Please note that figures must have been originally created at 300dpi or higher. Do not manually increase the resolution of your files. For instructions on how to properly obtain high quality images, please review our Figure Guidelines, with examples at: http://journals.plos.org/plosntds/s/figures

While revising your submission, please upload your figure files to the Preflight Analysis and Conversion Engine (PACE) digital diagnostic tool, https://pacev2.apexcovantage.com/ PACE helps ensure that figures meet PLOS requirements. To use PACE, you must first register as a user. Then, login and navigate to the UPLOAD tab, where you will find detailed instructions on how to use the tool. If you encounter any issues or have any questions when using PACE, please email us at figures@plos.org.

We hope to receive your revised manuscript by Nov 19 2019 11:59PM. If you anticipate any delay in its return, we ask that you let us know the expected resubmission date by replying to this email.

To submit your revised files, please log in to https://www.editorialmanager.com/pntd/

Sincerely,

Maria Victoria Periago

Associate Editor

Aaron Jex

Deputy Editor

Reviewer's Responses to Questions

**Key Review Criteria Required for Acceptance?**

**Methods**

-Are the objectives of the study clearly articulated with a clear testable hypothesis stated?

-Is the study design appropriate to address the stated objectives?

-Is the population clearly described and appropriate for the hypothesis being tested?

-Is the sample size sufficient to ensure adequate power to address the hypothesis being tested?

-Were correct statistical analysis used to support conclusions?

-Are there concerns about ethical or regulatory requirements being met?

Reviewer #1: The methods are appropriate and are clearly presented. My only minor concern is that previous work has been done on BZ binding to membranes; were these methods designed de novo or were they based on earlier, similar work? If the latter is the case, that earlier work should be cited.

Reviewer #2: - Bioinformatics of LGICs: were homology-based searches also used to confirm missing channels? This would help in case fragmented gene models (or missing gene models) did not meet the transmembrane count filter.

- L126 typo (missing 'washed')

Reviewer #3: (No Response)

**Results**

-Does the analysis presented match the analysis plan?

-Are the results clearly and completely presented?

-Are the figures (Tables, Images) of sufficient quality for clarity?

Reviewer #1: The results are clearly and concisely presented and the data are convincing. The authors might note that previous bioinformatics searches had already shown that S. mansoni lacks GABAr sequences.

Reviewer #2: - Throughout the manuscript, 'selectivity' is profiled by measuring in vitro effects on schistosomes and specific interactions with mammalian GABAAR channels. There is the real possibility that selecting for chemicals that maintain 'parasite-selectivity' and diminished affinity for GABAAR channels is akin to selecting for activity against a different homologous parasite and host target set. Since the mode of action that drives this selectivity is unknown, this possibility cannot yet be ruled out. While not in the scope of this manuscript, I think it would be valuable to devote some text to discussing this possibility.

- Differences in recovery among imidazobenzodiazepines (Fig 4) are not well-explained. I would be curious to know whether higher concentrations of XliHe-II-048 and SH-I-055 imidazobenzodiazepines would have prevented recovery - suggesting this is a threshold effect.

Reviewer #3: (No Response)

**Conclusions**

-Are the conclusions supported by the data presented?

-Are the limitations of analysis clearly described?

-Do the authors discuss how these data can be helpful to advance our understanding of the topic under study?

-Is public health relevance addressed?

Reviewer #1: The conclusions are reasonable and well-supported by the data. The sentence beginning on line 286 is unclear; it seems to indicate that it is expected that the sedative dose of MCLZ is the same as the therapeutic dose for schistosomiasis. Please clarify.

Reviewer #2: (No Response)

Reviewer #3: (No Response)

**Editorial and Data Presentation Modifications?**

Reviewer #1: Minor concerns only:

1.Line 55: current estimates of death due to schistosomiasis are much lower than 300,000; the cited refreence is 16 years old and should be updated.

2. Line 197: replace 'at' with 'for'

3. Use 'concentration' instead of 'dose' when referring to units of mass/units of volume

4. Line 264: delete 'exhibit'

5. Line 352: 'pose' is an unusual term in this context; replace with 'pocket'?

6. Line 375: delete first use of 'the'

Reviewer #2: (No Response)

Reviewer #3: (No Response)

**Summary and General Comments**

Reviewer #1: This is an interesting and important paper, helping to clear up a long-standing mystery in this field. I congratulate the authors.

Reviewer #2: This is a very interesting manuscript that seeks to re-explore the utility of benzodiazepines as agents of schistosome control. The authors make a compelling case that the dose-limiting sedatory effects of meclonazepam (MCLZ) are not necessarily coupled to MCLZ schistocidal activity. They make this case in two primary ways. First, bioinformatics analyses are used to show loss of GABAAR-like channels in this flatworm lineage and to suggest that different (yet unknown) target(s) exists for MCLZ in schistosomes. Second, various non-sedating benzodiazepines are shown to have strong effects on worm motility and tegument integrity. This study is a starting point to identifying benzodiazopines that fit this selectivity profile and identifying the likely novel mediators of benzodiazopine activity against schistosome parasites. I suspect that target identification will allow for broader screening efforts to identify other classes of chemicals that phenocopy these effects. The experiments were well-designed and presented. My comments are minor.

Reviewer #3: I enjoyed reading this paper and consider it to be of high interest to the field of parasitology. I think it is an excellent idea to reconsider the benzodiazepines as potential antischistosomal drugs, and exciting that the authors have found novel benzodiazepine compounds that appear to lack sedative side effects. 

I am not an expert in the area of protein-ligand docking so do not consider myself qualified to assess that section.

I recommend that this paper be accepted for publication, as long as the authors address several points:

- Figure 1: in figure 1 the authors show the GABA_A_R class is absent from Schmidtea, Echinococcus and Schistosoma. Given that there are many more flatworm species available in WormBase ParaSite, would it be possible for the authors to screen those for homologs too, to be absolutely sure these genes are not present in some trematodes? 

- Figure 2: In figure 2 the authors show 180 benzodiazepines that they screened. It's not clear to me how these compounds were chosen, are they just all benzodiazepines that the authors could get their hands on, or was there some criteria for selecting the compounds? Please could the authors clarify this.

- Figure 4: the authors show what happens with MCLZ at 5 microM but Xhe-II-048 at 10 microM. I may have missed it, but is there no effect with Xhe-II-048 at 5 microM? Also, I am wondering have the authors tested Xhe-II-048 on other species such as S. haematobium and tapeworms to see if there is an effect on tegument of other flatworm species, which might suggest it would act as a drug for those species too? 

- Discussion: It would be of great interest to find a drug that can be used for all the different species of schistosome, for example, for Schistosoma haematobium and S. japonicum also. Have the authors tested or considered testing the imidazobenzodiazepines against other schistosome species (and even other flatworm parasites like Fasciola and Echinococcus)? I presume that this will be a next step for the authors, in addition to tests on different stages of schistosomes (somules, juveniles), and in vivo studies with these drugs in mice? I would like the authors to add some discussion on what they consider the most important next step to develop these compounds as new drugs.

--- Avril Coghlan

PLOS authors have the option to publish the peer review history of their article (what does this mean?). If published, this will include your full peer review and any attached files.

Reviewer #1: No

Reviewer #2: No

Reviewer #3: Yes: Avril Coghlan

---

## [Editor Report · Decision Letter 1]

3 Oct 2019

Dear Dr. Chan,

We are pleased to inform you that your manuscript, "Non-sedating benzodiazepines cause paralysis and tissue damage in the parasitic blood fluke Schistosoma mansoni", has been editorially accepted for publication at PLOS Neglected Tropical Diseases.

Before your manuscript can be formally accepted and sent to production you will need to complete our formatting changes, which you will receive in a follow up email. Please note: your manuscript will not be scheduled for publication until you have made the required changes.

IMPORTANT NOTES

* Copyediting and Author Proofs: To ensure prompt publication, your manuscript will NOT be subject to detailed copyediting and you will NOT receive a typeset proof for review. The corresponding author will have one final opportunity to correct any errors when sent the requests mentioned above. Please review this version of your manuscript for any errors.

* If you or your institution will be preparing press materials for this manuscript, please inform our press team in advance at plosntds@plos.org. If you need to know your paper's publication date for media purposes, you must coordinate with our press team, and your manuscript will remain under a strict press embargo until the publication date and time. PLOS NTDs may choose to issue a press release for your article. If there is anything that the journal should know, please get in touch.

*Now that your manuscript has been provisionally accepted, please log into EM and update your profile. Go to http://www.editorialmanager.com/pntd, log in, and click on the "Update My Information" link at the top of the page. Please update your user information to ensure an efficient production and billing process.

*Note to LaTeX users only - Our staff will ask you to upload a TEX file in addition to the PDF before the paper can be sent to typesetting, so please carefully review our Latex Guidelines [http://www.plosntds.org/static/latexGuidelines.action] in the meantime.

Best regards,

Maria Victoria Periago

Deputy Editor

Aaron Jex

Deputy Editor

---

## [Editor Report · Acceptance letter]

8 Nov 2019

Dear Dr. Chan,

We are delighted to inform you that your manuscript, "Non-sedating benzodiazepines cause paralysis and tissue damage in the parasitic blood fluke *Schistosoma mansoni*," has been formally accepted for publication in PLOS Neglected Tropical Diseases.

Best regards,

Serap Aksoy

Editor-in-Chief

Shaden Kamhawi

Editor-in-Chief
